# Dietary Quality of Women of Reproductive Age in Low-Income Settings: A Cross-Sectional Study in Kyrgyzstan

**DOI:** 10.3390/nu14020289

**Published:** 2022-01-11

**Authors:** Aiperi Otunchieva, Jamila Smanalieva, Angelika Ploeger

**Affiliations:** 1Specialized Partnerships in Sustainable Food Systems and Food Sovereignty, Faculty of Organic Agricultural Sciences, University of Kassel, 37213 Witzenhausen, Germany; a.ploeger@uni-kassel.de; 2Department of Food Productions Technology, Faculty of Technology, Kyrgyz State Technical University after I. Razzakov, pr. Aytmatov 66, Bishkek 720044, Kyrgyzstan; jamila.smanalieva@gmail.com

**Keywords:** dietary diversity, nutrient adequacy ratio, women of reproductive age, the triple burden of malnutrition

## Abstract

Dietary diversity and adequate nutrient intake are essential for conducting a healthy life. However, women in low-income settings often face difficulties in ensuring dietary quality. This research assessed relationships between the dietary diversity, nutrient adequacy, and socio-economic factors among women of reproductive age (WRA) in Kyrgyzstan. A cross-sectional study was undertaken in four locations, including two rural and two urban areas in the north and south of Kyrgyzstan. A survey with pre-coded and open-ended questions was employed during the interviews of 423 WRAs aged 18–49. Data collection was conducted in March–May 2021. The average value body mass index (BMI) of WRA was 24.2 ± 4.6 kg/m^2^. The dietary diversity score (DDS) was higher among rural women (common language effect size) cles = 0.67, adjusted *p* < 0.001) in the northern region (cles = 0.61, *p* < 0.05) who have cropland (cles = 0.60, *p* < 0.001) and a farm animal (cles = 0.60, *p* < 0.05). Mean nutrient adequacy ratio (NAR) was below 1 in most micronutrients, whereas thiamine, riboflavin, vitamins B6 and C, folic acid, calcium, and magnesium were even lower than 0.5. Women with a kitchen garden or a cropland had better NAR energy (cles = 0.57), NAR carbohydrate (cles = 0.60), NAR fiber (cles = 0.60), NAR vitamin B1 (cles = 0.53), and NAR folic acid (cles = 0.54). Respondents who receive remittances and a farm animal have better NARs for energy, carbohydrates, fiber, vitamin B1, folic acid, iron, zinc, and mean adequacy ratio for 16 nutrients (MAR 16) than those who do not. Education and income have a negative correlation with dietary quality. This study contributes to the limited literature on the quality of diets in Kyrgyzstan. Hidden hunger and undernutrition are a severe problem among WRA in low-income settings. Recommendations are including study programs in nutrition, teaching households farming practices, and raising awareness on adequate nutrition.

## 1. Introduction

Around three billion people in the world could not afford healthy diets in 2019 [1]. The term coined ‘triple burden of malnutrition’ refers to a situation when overnutrition, undernutrition, and micronutrient deficiency exist. It is a serious public health problem in low-income countries [2,3,4], including Kyrgyzstan. Micronutrient deficiency is a lack of vitamins and minerals crucial to health, and it may result from poor diet and diseases [3]. Consumption of low-nutrient and energy-rich diets may lead to obesity when vital micronutrients are still lacking in human organisms [5]. Issues of obesity are relevant not only in the developed world, but also in developing countries that suffer from excessive energy intake leading to some non-communicable diseases [2].

High food prices and income inequality hinder affordability of diversified foods [1]. Household members may experience hidden hunger regardless of its financial situation. Hidden hunger is a silent but alarming issue affecting both disadvantaged and well-off families [6]. Still, maintaining dietary diversity is challenging for the poorest [7]. Low-income communities residing in the outskirts of urban areas are especially prone to lack of food security [8]. They tend to have poor and monotonous diets. Dietary diversity depends on household income and energy availability [9]. Shortage of low-energy nutrition may likely occur during lean season or food shortage periods [10].

Due to low incomes, high food prices, and lack of information and knowledge, women in disadvantaged households have low-nutrient diets, which silently lead to hidden hunger. Food diversity is important for adequate nutrition and health. The intake of diversified foods is often advised to prevent the body from lacking important micronutrients, vitamins, and minerals. Nutritious food refers to the sufficient intake of calories and quality food (variety and micronutrient value of food). However, although food security is a prerequisite for nutrition security, it is not enough for ensuring the sufficiency of maintaining good nutritional status [11]. Consuming a variety of foods from different food groups is the underlying method for achieving and preserving nutrient adequacy [12].

Women in low-income settings are especially vulnerable to undernutrition. The intake of three minerals—iron, iodine, and calcium—is vital for woman of reproductive age. Proper nutrition of women, especially during pregnancy, is vital for her and her future offspring’s brain and neurodevelopment [13]. Iodine is vital for brain development. Adequate intake of iodine in pregnancy is important, as in-utero deficiency may have lifelong consequences for the offspring [14]. The WHO iodine recommendation for pregnant women is 250 μg/d (in comparison to the German Nutrition Society’s (DGE) recommendation for pregnant and breastfeeding women, which is 200 μg/d), which is higher than the 150 μg/d recommendation for adults. Adequate nutrition, including the intake of diversified foods, supports the organism by fulfilling important macro- and micro-elements. Vitamins and minerals have diverse important biochemical functions in cellular processes of human organisms. For example, vitamin A acts as a regulator of cell and tissue growth and differentiation. Vitamin D is essential for bones and other organs as it regulates mineral metabolism. Important in reproductive age, vitamins C and E act as antioxidants, a substance that neutralizes a free radical, thereby preventing oxidative injury to cells and tissues [15,16].

Iron is an essential part of hemoglobin for oxygen transport and of myoglobin for transporting and storing oxygen in the muscle and releasing it when needed during muscle contraction [17]. The recommended dietary allowance (RDA) value for Fe is 15 mg and 30 mg for pregnant women [18]. Calcium is important for the development of bone, teeth, and muscles and is associated with vitamin D metabolism as well [17]. The RDA for Ca is 1000 mg (DGE, 2021). Selenium has antioxidant and anti-inflammatory properties [19]. This article explores 16 macro- and micronutrients, but the above-mentioned micronutrients are also vital for women of reproductive age.

### Literature Review on Nutrition in Kyrgyzstan

Kyrgyzstan is a lower-middle-income country [20], where nutrition-related issues exist that impede overall human development. Changing lifestyles coupled with low physical activity and consumption of processed foods and beverages in developing countries causes issues such as becoming overweight [21]. The nutrition transition is taking place in the Kyrgyz society as well. Around 10–15 years ago, Kyrgyzstan’s population had higher undernutrition and lower overnutrition levels among European and Central Asian countries [22]. However, this picture is changing as becoming overweight becomes a severe problem for many people in the country. Additionally, 80% of deaths in 2019 were due to non-communicable diseases (NCDs) [23]. An analysis of the diseases model shows that, for the last 30 years, infectious diseases have been decreasing whereas proportion of people with non-communicable illnesses (NCDs) has been increasing [24].

The National Food Security and Nutrition Program, approved by the government of the Kyrgyz Republic Decree Nr. 618 dated 4 September 2015, points out, among others, the importance of a balanced diet in addressing micronutrient deficiency. Irrespective of this program, maternal health requires special attention in Kyrgyzstan. Despite healthcare reforms, maternal mortality remains high [25]. For the period of 1990–2015, it reduced only by 7%. In Kyrgyzstan, a high percentage of anemia, iodine deficiency, and diabetes among pregnant women is common [26]. The prevalence of anemia among pregnant women was 39.8% in 2016 [27], whereas most of the anemia among WRA was 36.2% in 2016 [28]. The prevalence of overweight among women was 48.8% in 2016, and the prevalence of obesity among women constituted 18.6%. These indicators steadily increased since 1975 (27.1% and 5.6%, respectively, in 1975) [29].

The nutritional status of women is of special concern. Nationally, over half of women have a normal body mass index (BMI) (57%), while 7% are undernourished or thin (BMI less than 18.5), and 36% are overweight or obese (BMI 25 or higher) in the country. The mean BMI for women aged 15–49 is 24.1, which falls in the standard BMI classification. Differences in BMI levels by background characteristics are apparent. Women aged 15–19 are more likely than women in other age groups to be thin or underweight (18% versus 2–8%). In contrast, the proportion of overweight women increases with age; among women aged 40–49, 41% are overweight, and 31% are obese. Rural women are more likely to be overweight or obese than urban women (38 and 32%, respectively). By region, the proportion of undernourished women does not vary much; however, the ratio of overweight women ranges from 19% in Bishkek to 30% in the Naryn region. Obesity is more common among women from the Issyk-Kul, Osh Oblast, Talas, and Chui regions (14–16%) than among women from other areas (8–10%). Women with primary general education are more likely to be thin and less likely to be overweight or obese than women with more education. Similarly, women in the highest wealth quintile are slightly more likely to be thin and less likely to be overweight or obese than women from less affluent households. Among women, anemia, iron deficiency, and lack of vitamin A are widespread micronutrient deficiencies in Kyrgyzstan [30].

Improving dietary diversity is a strategy that can be employed on a micro-level. Its assessment can be performed though calculating the number of different food groups consumed over a specific reference period [31]. On a global scale, literature on dietary diversity in Africa [10,32,33,34,35,36,37] is abundant. DD studies in South-East Asia [38,39] and India [40,41] enrich the literature on this topic as well. These works consider different socio-economic factors, including farm production and market access together with other determinants, and their exciting insights are related to the dietary diversity of women, children, and households. Staples are the commonly consumed food group in all these areas. Legumes, green leaves, vegetables, fish, and roots and tubers seem to be widely consumed in Africa. Meat and poultry are less commonly consumed food groups in some African countries.

The nutrition topic is understudied in the Central Asian region. Existing literature on maternal nutrition in Kyrgyzstan, among other things, is insufficient [25]. Some nutritional indicators made substantial improvements. For the period 1997–2004, child stunting prevalence reduced nationally from 36% to 13%. Although, on the province level, the decline rates have some differences [42,43]. Despite these advancements, maternal nutritional status did not make rapid improvements. This issue requires further research and policy analysis [42].

Different methodologies for measuring food diversity exist in the literature. Some indexes are based on the quantitative distribution of consumed foods as a measurement for diet diversity [44]. Some of the main measurements are dietary diversity score (DDS), food variety score (FVS) [10], Household Dietary Diversity Score (HDDS) [45], and Minimum Dietary Diversity for Women of Reproductive Age (MDD-W) [46]. FVS, DDS, DDS half, and dietary serving score were employed by [47] as a method to assess dietary diversity among the elderly in Sri Lanka. Dietary diversity indicators group foods together when they are considered nutritionally similar and/or play the same role in the diet [31]. While developing the MDD-W, many different candidate indicators, with different numbers of food groups and different food group definitions, were considered. The indicator based on the ten groups described further in the methods part had a stronger relationship to micronutrient adequacy than other candidate indicators with different groupings [48]. Thus, this paper employs MDD-W (further referred to as DDS).

The study objectives are (1) to understand how diversified the diet of women 18–49 years old in Kyrgyzstan is and their similarities and differences in urban and rural areas; (2) to find out determining factors impacting dietary quality (DDS and nutrient adequacy) among women; (3) additionally, we intend to understand whether DDS can be a proxy for micronutrient deficiency in the context of WRA from low-income families of Kyrgyzstan.

This article intends to make the following contributions to the literature. Firstly, dietary quality assessment is performed on two bases: evaluation of dietary diversity and nutrient adequacy ratio and mean adequacy ratio. Most works rely on the simple count of food groups. DDS alone is not a proxy for measuring food security. Ensuring that individuals or households have nutrient adequacy requires more data for analysis. Most articles that study DD do not consider nutritional outcomes [32,34,38,41]. A simple count of food groups does not completely picture nutrient sufficiency [33]. High DD does not always translate into the absence of micronutrient deficiency. Therefore, our study intends to fill the existing gap in the literature and employs nutrient adequacy ratio and mean nutrient adequacy ratio to better picture the diet quality of women in Kyrgyzstan.

Second, dietary diversity is believed to be a plausible indicator for assessing micronutrient deficiency, although it has its peculiarities [49]. Determination of micronutrient adequacy is expensive if one analyzes it through blood [49]. In our article, we calculated micronutrient deficiency for each respondent separately. We believe this approach gives us more accuracy in understanding hidden hunger among women of reproductive age. Third, understanding how households with limited resources nurture themselves and comprehending their diet composition will give insights to policymakers to improve current nutrition strategies and programs. To the best of our knowledge, dietary quality assessment using a scientific approach has not been performed in Kyrgyzstan yet.

Based on the existing literature, our hypothesis states that, as the society in Kyrgyzstan undergoes nutrition transition, women suffer from overnutrition, undernutrition, and micronutrient deficiency in low-income settings. Due to the poor dietary diversity, women are short of vital micronutrients. Socio-economic, demographic, markets, and farms are the determining factors impacting WRA’s diet quality.

## 2. Materials and Methods

The study has a community-based cross-sectional survey design, and data were collected from women of the 18–49 age group (*n* = 423) between March and May 2021. Random and snowball sampling was followed in the selection of women in Kyrgyzstan. Respondents were selected in the two biggest cities of Kyrgyzstan: Bishkek (*n* = 99) and Osh (*n* = 98), as well as two rural areas of At Bashy (*n* = 90) and Aravan (*n* = 136) districts. The survey included women residing in disadvantaged neighborhoods, including family dormitories and slums near big markets (Dordoi, Osh bazar). Pregnant, lactating women and those from well-off families were excluded from the survey.

The survey was based on pre-coded and open-ended questions. Methods include (1) descriptive statistics of cross-sectional data; (2) analysis of demographic and socio-economic variables with DDS (dietary diversity score), NAR (nutrient adequacy ratio), and MAR (mean adequacy ratio) for 16 nutrients, which are energy, protein, carbohydrates, fat, fiber, vitamin A, vitamin C, vitamin E, vitamin B6, folic acid, iron, zinc, magnesium, calcium, riboflavin, and thiamine.

### 2.1. Study Area

The field research was conducted in different cultural, geographical, agro-ecological, and historical locations (Table 1). Two urban areas, Bishkek (capital) and Osh (the second largest city), and two rural areas (At Bashy and Aravan districts) were the study locations. At-Bashy district of Naryn oblast is situated on both banks of the Naryn River (one of the central headwaters of the Syrdarya) [50]. It is a mountainous region whose economy depends predominantly on animal husbandry. It is located at 2044 m above sea level and consists of 98% of ethnic Kyrgyz.

The Aravan district Osh oblast is located in one of the world’s most populous places, Fergana Valley (Figure 1). It is an ethnically mixed location comprising Uzbeks, Kyrgyz, Russians, Tajiks, and others near Uzbekistan’s border. It is at 963 m above sea level. Agriculture is a historically significant part here, including planting cotton, fruits and vegetables, nuts, and many other crops, which are later sold in cities.

### 2.2. Questionnaire

The survey included modules on socioeconomics, demographics, farm, market, and 24-hour diet recall (Figure 2) and was administered by female trained encoders. Determinants of nutrition include distance to the market, household income, and household production [4]. Some studies found that cropland and kitchen garden positively impact dietary diversity score [40]. Therefore, we added these variables to our questionnaire.

Interviews were conducted in official languages of the Kyrgyz Republic, which are Kyrgyz and Russian. The interview data were collected with the online tool Typeform [52], transferred to the descriptive Excel datasheet, and then transferred to the quantitative statistical analysis and programming language R. A 24-hour diet recall was conducted based on 16 food groups and was then grouped into ten food groups developed by FAO [46] to assess dietary diversity. Additionally, the calculation of macro- and micronutrients was performed with the software Nutrisurvey (EBISpro: Willstätt, Germany, 2007) [53]. The draft questionnaire was piloted with 25 people. After the comments of test respondents, changes to the questionnaire were made, such as rephrasing the questions, changing to local names, and adding more local foods.

### 2.3. Anthropometric Measurements

Enumerators were trained by the medical personnel to perform anthropometric measurements. Each respondent was weighed (in kg) by the mobile scale and measured by the mobile stadiometer (in cm). BMI is an indirect measure of body fat through height and weight, and it is calculated by kg/m^2^. Four categories of BMI exist—underweight (<18.5), normal weight (18.5–24.9), overweight (25–29.9), and obese (above 30) [29].

### 2.4. Nutrient Adequacy Ratio

For the investigation of dietary quality of WRAs, two methods were used in this study. Firstly, a 24-hour diet recall was based on the internationally accepted methodology for assessing women’s macro- and micronutrient deficiency. We recorded all foods, including dishes, together with exact ingredients and portion sizes for precise calculation for portions that were consumed for breakfast, lunch, dinner, and snacks in between. The entry of ethnic dishes’ recipe into the software was according to the book “Kyrgyz Modern Cuisine” [54].

The nutrient adequacy ratio (NAR) is calculated for each nutrient by using the Equation (1).
(1)$$\mathrm{NAR}\;=\frac{\mathrm{Actual}\;\mathrm{intake}}{\mathrm{Recommended}\;\mathrm{daily}\;\mathrm{intake}}$$
(2)$$\mathrm{MAR}=\frac{\mathrm{Sum}\;\mathrm{of}\;\mathrm{NAR}}{\mathrm{Number}\;\mathrm{of}\;\mathrm{nutrients}}$$

Due to the lack of recommendation for dietary intake (RDI) developed for women in Kyrgyzstan, we employed the RDI used by Russia and Germany. The mean nutrient adequacy ratio (MAR) is calculated for 16 macro-and micronutrients using the equation (2). When NAR = 1 and MAR = 1, it indicates that nutrient intakes have met the RDI. NAR and MAR> or <1 means lack or overconsumption of specific micronutrients [55].

### 2.5. Dietary Diversity Score

DDS consists of ten main following food groups: 

1. Grains, white roots and tubers, and plantains;

2. Pulses (beans, peas and lentils);

3. Nuts and seeds;

4. Milk and milk products;

5. Meat, poultry, and fish;

6. Eggs;

7. Dark green leafy vegetables;

8. Other vitamin A-rich fruits and vegetables; 

9. Other vegetables;

10. Other fruits.

Women who consumed less than five items out of the ten food groups were more likely to be micronutrient poor. Those who had more than five items out of the ten food groups were less likely to be micronutrient poor [46]. We asked which ingredients were used and registered all different elements included in the meal (whenever the quantity was >15 g per serving). On the spot, we recorded per meal which foods and portions had been consumed by ticking the related food group from, for example, the list of 16 different food groups shown in the questionnaire and used 10 groups in this paper [46]. Pictures of a plate, a teacup, a soup cup, a tablespoon, and a teaspoon were employed for identification of portions.

### 2.6. Data Analysis

Statistical analysis was conducted using the R language for statistical computing [56] and RStudio [57]. Illustrations were built with Excel. Continuous variables were presented with their respective mean, standard deviation (SD), standard error (SE), median, and first (Q1) and third (Q3) quartiles. Medians were calculated, and averages were shown. Categorical variables were presented with proportions and percentages. Comparisons were carried out using Mann–Whitney U tests for continuous variables. The threshold used for statistical significance was *p* < 0.05.

Effect sizes were reported as common language effect size (using package “canprot” [58], pseudo-median, and its nonparametric 95% confidence interval (CI).

Correlations were assessed using Kendall’s tau. The Benjamini–Hochberg method was used to control the false discovery rate [59].

## 3. Results

### 3.1. Socio-Economic Characteristics of WRAs

The socio-economic characteristics of respondents are summarized in Table 2. The sample size involved 423 WRA, 226 women from rural areas and 197 women from urban areas, with an age range of 18–49. Around two-thirds of respondents (*n* = 287) were Kyrgyz and 136 were representatives of other ethnic groups such as Dungans, Koreans, Russians, Tajiks, Uigurs, and Uzbeks. Most of the respondents (*n* = 170) were literate, having secondary school diplomas and higher education certificates (*n* = 167). Of the respondents, 73% of the women were married, and 64% lived with at least one child under 18 in one household. Most households (59.6%) were considered as large families consisting of more than four members, predominantly residing in rural areas (75.7%). More than half of the women’s main income source was employment. Agriculture was the second main income source for almost 17% of respondents, out of which the majority resided in rural areas. An important difference between women in rural and urban areas is that most of the women’s source of income in rural areas originates in agricultural activities (28%) than in urban areas (3%). Remittances are vital for around 18% of rural and 22% of urban women. More than a third of respondents’ household income was (41.8%) 0–10,000 Kyrgyz soms, which is roughly 0-$115 US.

### 3.2. Nutritional Status

The mean height and weight of women in our sample were 161 cm (±6.6 cm) and 63 kg (±12.6 kg), respectively. The mean BMI of respondents was 24.2 ± 4.6 kg/m^2^, close to the threshold of 25.0. Although 58.8% of women had a normal BMI, more than one-third of survey participants were overweight or obese (36.6%), and undernutrition was among 7.6% of participants.

### 3.3. Individual Food Consumption and Diet Diversity Score (DDS)

Almost all women reported consumption of starchy staples (Figure 3). The second most consumed food group was meat, offals, and processed meat products. Consumption of fish and fish products was minimal. Vitamin A-rich vegetables and fruits followed as the third most eaten food group among women in Kyrgyzstan. Traditional cuisine consisted of different dishes, which predominantly included staples, meat, and vitamin A-rich vegetables. Widely consumed were *besh barmak* (noodle, meat, onion), *plov* (rice, meat, carrots, onion), *lagman* (noodles, meat, carrot, red pepper, and other vegetables), *manty* and *oromo* (noodle, meat, onion, pumpkin/potato), *grechka* (buckwheat, meat, carrots), *pelmeni* (noodles, potato, meat, onion), *pirojki* (dough, potato, onion), *kesme* (noodles, meat, onion, carrots), and *mastava* (rice, meat, onion, carrot). Milk and milk products were the fourth most important in the diet of women in Kyrgyzstan. Almost 77% of rural women reported consuming dairy products in the last 24 hours when the survey was conducted. Nuts and seeds, beans and peas, and eggs were consumed moderately, constituting more than 30% of rural women and much less among women in the cities (14.72%, 14.21% and 26.4%, respectively). Pea soups were consumed by some respondents, whereas nuts and seeds were mostly as additions to salads, sweets, and dishes. Fried eggs were favored due to their quick preparation. Vitamin A-rich dark green leafy vegetables were eaten by a very small number of women comprising around 30% in both locations, although dill, coriander, parsley, green garlic, and chives usually grow in the spring period when data collection was conducted (Figure 3).

Generally, fruits and vegetables were consumed by few women, 21.68% and 6.64% in rural areas and 8.63% and 2.54% in cities, respectively. Seasonal availability of fruits and their high prices off-season might hinder women from low-income families to afford them. There is a general trend in all food groups that their consumption was higher among rural women than in urban areas. However, DDS is still quite close to the threshold 5, meaning that there is little difference between women in two different geographical settings. Relatively low mean DDS in both areas may likely be because the data collection took place during the lean season when food availability was low and their prices were high.

The division into food groups is an internationally recognized method. This study used the updated recommendations developed by FAO (2021), including ten food groups (Figure 3). Locally consumed foods included both traditional Kyrgyz as well as those that originated in other cultures. Due to past migration processes, local cuisine was influenced by the diet of ethnic groups such as Koreans, Dungans, Uigurs, Russians, Uzbeks, and others. Besides noodles and rice, starch staples include buckwheat, barley, potatoes, and others. Meat foods include those indicated in Table 3 and very little fish. Due to the past nomadic food culture and current semi-nomadic lifestyle in some rural areas, dairies are an essential cuisine element. Among others, these foods include homemade ones such as kurut (dried concentrated ayran), suzmo (concentrated yogurt), kajmak (creamy milk residue), ayran (sour milk), sary maj (ghee), kymyz (fermented mare’s milk), and many others.

Consumption of animal-based proteins is typical. Meat, organ meat, and milk products belong to one of the most consumed food groups. Nearly 40% of households possessed at least one farm animal, primarily cows, sheep, goats, chickens, or horses. Almost 50% of respondents owned a kitchen garden or cropland where they grew numerous crops, fruits, and vegetables. In mountainous areas such as At Bashy, mostly potato, barley, beet, and garlic are grown. At a lower altitude valley area in Aravan, a variety of plant foods is broad, including cucumber, tomato, radish, green onion, red pepper, eggplant, cabbage, chives, dill, strawberry, persimmon, grape, fig, peach, pomegranates, cherry, apricot, apple, rice, corn, pistachios, and lemon.

### 3.4. Nutrient Intake and Diet Quality

Currently, Kyrgyzstan lacks a state-confirmed recommended daily intake (RDI) for women between 18 and 49. Therefore, we used RDI from the State Sanitary and Epidemiological Regulation of the Russian Federation (2009) and the DGE German Nutrition Society (2021) in this study. Both RDIs have slight differences, for example, RDI for protein by the Russian RDI is 65 g, whereas the DGE recommendation is based on weight 0.8 to 1 g/kg. Most norms for vitamins and minerals in the Russian RDI are higher than in the German one (Table 4).

Diet analysis was performed to derive nutrient intakes of respondents. Total mean energy intake was 1647 kcal, whereas protein, fat, carbohydrates, and fiber were around 59 g, 61 g, 211 g, and 16 g, respectively. These indicators are lower than the Russian RDI. According to the recommendations by the DGE, some mean NAR indicators show around 1, meaning that respondents have adequate nutrient intake daily. These nutrients include protein, vitamin A, E, and zinc, but, according to the Russian RDI, all macro- and micronutrients are either below or close to 1.

Regardless of these norm differences between these two countries, mean NAR is below 1 in most macro-and micronutrients when calculated with RDIs, some of them being even lower than 0.5. These are the critical micronutrients such as vitamins B1, B2, B6, and C and folic acid, calcium, and magnesium. These results indicate that hidden hunger exists, and undernutrition is a severe problem among WRA in low-income settings in Kyrgyzstan.

#### Analysis of Dietary Diversity Concerning Some Socio-Economic and Farm Variables

We employed Mann-Whitney U tests to carry out group comparisons for continuous variables. Namely, we performed comparisons for DDS with socio-economic and demographic variables (Appendix A). We found that differences exist in residence, region, kitchen garden or cropland, and farm animals in terms of dietary diversity (Figure 4). When a respondent in the rural group was compared to the respondent in the urban group, the respondent from the urban group had higher DDS than the respondent in the rural group in only 33% of cases. This means that in 67% of cases, DDS was higher among rural WRAs than in urban. In 61% of cases, DDS in the north was higher than in south. In 39% of cases, DDS in the south was more than in the north. This is an interesting result because the south is traditionally more favorable for growing fruits, vegetables, pistachios, and almonds due to its warm climatic conditions. Considering these results, knowledge on nutrition, significance of dietary diversity, food storage, and gardening should be improved in the southern region.

In 59.9% of cases, DDS among women with a kitchen garden or cropland was higher than women without a kitchen garden or cropland. Women who have a least one farm animal also had better DDS in almost 60% of cases. These results indicate that women have more diversified diets in rural areas, in the northern region, and those who have cropland or a kitchen garden and at least one farm animal in comparison to those who live in cities in the south and do not possess a piece of land and a farm animal.

### 3.5. Nutrient Adequacy Ratio and Its Socio-Economic Factors

We employed Kendall tau to analyze the correlation between NAR for 16 nutrients with income, education, and BMI. During our analysis, we discovered that a correlation between income and some NARs existed. Namely, there is a very weak negative statistically significant correlation between income and NAR for carbohydrates, fiber, vitamin B1, folic acid, and magnesium (Figure 5). This indicates that the more households earn, the less of these micronutrients’ women obtain from their diets. However, when corrected for multiple testing, the statistical significance for carbohydrates, vitamin B1, and magnesium was no longer observed.

The situation with education also indicates a similar trend (Figure 6). There is a very weak statistically significant negative correlation between education and NAR energy, carbohydrates, fiber, vitamin E and B1, folic acid, calcium, magnesium, iron, and Mar-16. The statistically significant difference persists even when corrected for multiple comparisons, except for NAR, vitamin E, and calcium. We conclude here that the higher the level of education of a woman, the less of the nutrients she obtained. BMI and NAR fiber have a statistically significant very weak positive correlation (Figure 7).

Residence of a respondent matters when it comes to nutrition. Our results show that when a respondent in the rural group was compared to the respondent in the urban group, the respondent from the urban group had higher NAR energy than the respondent in the rural group in only 41.7% of cases. In 58.3% of cases, women from rural areas had higher NAR energy than those in urban areas (Figure 8). When we compare NARs in urban and rural areas, we see that rural areas have better NARs for some macro- and micronutrients than those women residing in urban areas. This trend applies to NAR carbohydrates (urban for 0.36 and rural for 0.63), fiber (urban had 0.38 of cases and rural had 0.62), folic acid (urban was 0.43, rural was 0.57), calcium (0.43 in urban and 0.57 in rural), and magnesium (0.45 in urban and 0.55 in rural). All these results are statistically significant.

We also compared women with a kitchen garden or cropland and found out that those who had them have better NARs for some macro- and micronutrients. In 42.9% of cases, women without a plot of land had better NAR for energy than those with plot of land. This means that 57.1% of women with plot of land had higher NAR energy than those without one (Figure 9). WRA with a kitchen garden or a cropland had better NAR carbohydrate (0.60), NAR fiber (0.60), NAR vitamin B1 (0.53), NAR folic acid (0.54), NAR magnesium (0.54), and Mar-16 (0.54) than women without it.

Store distance is essential, as those respondents who live closer to the food market (less than 2 km) have better NARs for vitamin B2 and B6 (Appendix A). Remittances play a significant role in ensuring nutrient adequacy. When comparing a group receiving remittances and a group not receiving remittances, we found out that the former had better NARs for carbohydrates (0.56), fiber (0.57), vitamin B1 (0.56), folic acid (0.56), iron (0.56), zinc (0.55), and Mar-16 (0.54) (Figure 10). Women who have at least one farm animal had better NAR carbohydrate (0.63), NAR fiber (0.63), NAR energy (0.58), and NAR folic acid (0.54). These results are also statistically significant after multiple comparisons (Figure 11).

## 4. Discussion

The study aimed to understand diet diversification, nutrient adequacy, and its determining factors among women between ages 18 and 49 residing in low-income communities in Kyrgyzstan. Analysis showed that women in rural areas have more diversified diets than those living in cities. Consumption of simple carbohydrates (white rice, wheat noodles, white bread) along with traditional foods based on meat and milk products was popular among the respondents. Intake of plant-based protein (legumes, nuts, pulses) together with other fruits and vegetables was quite low. Assessment of nutrient adequacy for 16 nutrients showed that women in our samples do not obtain sufficient micronutrients, minerals, and vitamins from their daily diet. Shortage persists especially in iron, magnesium, calcium, folic acid, and vitamins C, B1, B2, and B6. 

Our results also indicate that nutrient adequacy is better in rural areas as well. Having a plot of land or kitchen garden and possession of a farm animal seem to positively impact nutrition of women. Women receiving remittances also have better NARs for carbohydrates, fiber, vitamin B1, folic acid, iron, zinc and Mar-16.

Malnutrition can be in many forms, including hunger, obesity, and micronutrient deficiency [60]. Our results are similar to many other studies [4,32,33,61] and showed that cereal-based nutrition is common among WRA. A woman’s diet in our sample was mainly based on noodles, bread, rice, buckwheat, potatoes, and other starchy foods. Meat and its homemade products have the most crucial role in nomadic cuisine. It is the main dish for guests served in any Kyrgyz home in different variations, including as an ingredient in salads. Meat is mostly served with noodles, the traditional food named *beshbarmak*. It is even believed that meat consumption makes one healthy, and children from a young age are given meat and animal fat. According to our results, meat, organ meat, and poultry was the second-highest reported food group. However, low-income families may have difficulties in buying meat due to its high price. We think its consumption might be lower than reported. Similar to in Afghanistan [7], milk consumption is high among Kyrgyzstani women. Nuts and seeds, lentils, and other fruits and vegetables constitute a little portion of most women’s diet in our sample.

### 4.1. Socio-Economic Factors and Nutrition Adequacy

For the last 20 years, Kyrgyzstan has overcome important nutrition challenges. It happened due to poverty reduction, remittances, health sector reforms, and the improved nutritional status of women and children. Several nutrition intervention programs were introduced in Kyrgyzstan, such as salt iodization, sprinkles, micronutrient powder, and food fortification [23,43].

Regardless of these improvements, Kyrgyzstan spent much fewer resources on maternal health in relation to child health [25]. Policies for improving child health and nutrition often overlook the role of a mother. Maternal health has been discussed in the context of child health due to the various intervention programs introduced to improve child nutrition. Resources have been so far directed to enhance child health while its lesser share has been allocated to the future mothers’ health and nutrition in Kyrgyzstan. Unfortunately it seems to be a global trend [62]. Some micronutrient determinants remain poor among WRA. Women with monotonous diets and low dietary diversity tend to have micronutrient deficiency [63].

Several studies show that education positively impacts DD [32]. Our analysis found no statistically significant correlation between DD and education of women and a very weak negative correlation between some NARs and education. These results also apply to income. Educated women tend to work and have higher payments. This might negatively impact her nutrition. Meal skipping was common among WRA in our sample, especially in urban areas. Working women have challenges organizing time for cooking, although they obtain more income than a housewife. Time was a limiting factor in another study as well [4]. Limited time for cooking might negatively impact nutrition, and this situation conversely impacts DD. Other studies also show that women’s education does not influence their diet quality [10]. Improving knowledge on proper nutrition should be performed [61]. Strategies for food preservation and its later consumption were common among households 30 years ago. Traditional cuisines of nomad food culture entail preservation methods such as drying, salting, and fermentation. Exploration of these methods and knowledge sharing could be one of the strategies for fighting malnutrition among WRA.

Empowered women tend to feed their children and consume diverse diets [38]. According to a study, education and advocacy positively impacted DD [61]. Nutrition knowledge and awareness are among the main determinants of DD [4]. Classes on nutrition and home economics should be an integral part of the school program [32]. A previous study indicated that [38] diverse food production is associated with diverse diets for children and women. These important aspects should be considered when developing food and nutrition policies.

Socio-economic determinants played a crucial role in reducing stunting and improving nutrition due to remittances sent by migrants. Migration is critical for enhancing household food security and diet diversification [42]. Migrant-sending communities gain benefits through receiving remittances by improving their nutritional status. A number of studies state that food security is positively associated with migration and remittances [64,65]. Cash transfers improve the dietary diversity of women [66]. Results of this study also show that remittances positively impact the nutrient adequacy of women. 

### 4.2. Farm Determinants and Market Access

A study in India showed that kitchen gardens positively influenced household members’ nutrition by increasing fruits and eggs intake [61]. Our results show that women with homestead gardens or plots of land have better DDS and better NARs for some macro- and micronutrients than those without one. It is an encouraging result, as our study also indicated that women residing in rural areas have a bit higher DDS than those living in cities. Almost all families own a plot of land (a few are rented) in villages in Kyrgyzstan. They tend to grow cash crops or fruits and vegetables. Although it may increase fruits and vegetables intake for households, the availability of kitchen gardens or cropland alone does not guarantee proper micronutrient intake [61]. Teaching farming techniques, efficient use of water, and marketing are essential skills for family members that need to be promoted. In addition to the availability of home gardens, agricultural training and nutrition counseling improves food security and dietary diversity [67].

Involvement in the food markets might positively or negatively impact DD. Some studies show that households engaging in food markets have more diversified diets [35] and market participation, and they need access to improve the nutritional outcomes of families. The close location of households to the market and the more they participate, the better DD they have [36]. Markets can also facilitate ensuring food diversity during the lean season [7]. Compared to other studies [4], our research showed that market distance and whether households sell agricultural products or consume themselves have no statistically significant association with DDS. Although our study did not consider the volume of food production, these results contradict the previous study conducted in Nigeria [34]. However, some micronutrient adequacy was higher among those women residing close to the food market in our sample. Further research on this factor and improving sampling might also show the above-mentioned results. However, we still know that market proximity might improve nutrient adequacy.

### 4.3. Urbanization

Diets tend to change with urbanization processes. Consumption of fast food, semi-manufactured food, and street foods is common among urban dwellers. Poor and inadequate diets may likely be a significant cause of non-communicable diseases. The results indicate a presence of hidden hunger, where nutritional status based on BMI shows either normal (more than 50% of respondents) or overweight/obese (more than one-third of respondents). Still, NAR values for all macronutrients and micronutrients (except for Vitamin A and Zinc) are below than 1. Worldwide, adult women tend to be more obese and overweight than men, leading to an increased risk during pregnancy and to reproductive health [68]. Despite numerous food supplementation and fortification programs, micronutrient deficiencies are still high in Kyrgyzstan. According to [69], Kyrgyzstan belongs to the countries with a high presence of trans fats in popular products. We omitted unhealthy food from the results section, as it does not belong to the FAO 2021 food group classification. However, their consumption (pizza, hamburger, soda drinks, shawarma, pirojki, belyashi) was also high, especially in urban areas.

Legumes, green leaves, and fish seem to be widely consumed in African and South-East Asian countries. Compared to these cultures, Kyrgyzstan consumes more animal-based protein such as meat and milk products and less plant-based protein such as pulses and nuts/seeds. Fish is also rarely consumed, likely since Kyrgyzstan is a landlocked country with no access to seashores. On the other hand, Kyrgyzstan has a handful of rivers and lakes, but fish consumption is still limited among low-income families.

Agricultural products are the principal sources of most nutrients [12]. Consumption of diverse foods is both healthy and environmentally friendly. Negligence of local food varieties leads to biodiversity loss. Although our results show that some traditional foods are widespread among women in our sample, important nutrient-dense foods and wild foods are missing in their diet. Incentivizing women to consume locally available nutrient-rich foods is a reasonable strategy instead of promoting micronutrient supplementation, which is hard to afford for the women in low income context [63].

DDS can partly be proxy for understanding micronutrient deficiency. However, assessment of nutrient adequacy gives us a better picture and more detailed information on which elements are lacking in human organisms. Our analysis showed that WRA do not obtain sufficient micronutrients from their daily diet.

### 4.4. Strengths and Limitations of this Study

This study has some strengths and limitations. Assessment of dietary diversity is performed with the calculation of intake of at least five foods from the FAO’s ten food groups. This method is quite useful, as it requires less resources. Our article went a step further to assess nutrient adequacy for 16 macro- and micronutrients to better understand the diet of women in poor communities. This paper attempted to fill the gap of the nutrition literature through making an overview of consumption habits among women in Kyrgyzstan. Understanding food culture of Central Asian nations alleviates the development of intervention programs directed at improving nutritional status of local women.

A 24-hour recall is a commonly used tool to assess dietary intake for target populations. However, it has its drawbacks, as it might not entirely reflect the usual food intake of a respondent. Due to time and financial limitations, it was convenient for us to employ a 24-hour dietary recall. With sufficient resources, it is also exemplary to use three or seven-day recalls. However, an advantage of the 24-hour diet recall is the absence of memory bias.

We included 16 nutrients in our study because these were analyzed with the software Nutrisurvey (EBISpro: Willstätt, Germany, 2007). Other vitamins and microelements are vital for women in reproductive age such as selenium, iodine, and others that are not part of this study.

Food security in rural areas depends on seasonality, which may impact micronutrient intake. Studies conducted in rural areas where food security depends on seasonality indicate that households had poorer dietary diversity due to high food prices and lower-income shortages [34]. Some longitudinal studies have shown that DDS is distinguished during growing and harvesting season. As this study was cross-sectional, we captured the nutrition picture during the growing season between March and May. This study was conducted in the lean season. Harvesting season might have shown better DD similar to [1]. A post-COVID-19 situation in the country also may likely affect women’s nutrition. The country’s seasonal food availability and devastated economic situation after the COVID-19 crisis worsened families’ status, which mostly lacked financial resistance. This might result in NAR being below 1 for most nutrients in our results. The exact effects of the pandemic on food security have not yet been assessed, but assumptions that stunting, wasting, and overweight rates might be higher exist [1].

## 5. Conclusions

Lack of food security and adequate nutrition remain to be public health problems among low-income households in Kyrgyzstan. Results of this study indicate that most WRAs consume starchy staples, flesh foods, and vitamin-A-rich vegetables. Meat and milk products are the sources of animal protein consumed by women. The calculation of mean NAR showed below 1 of energy, micronutrients such as vitamins B1, B2, B6, and C, folic acid, calcium, and magnesium and indicated malnutrition and hidden hunger. Despite favorable climatic conditions in Kyrgyzstan, fruits and vegetables are not widely used in the WRAs diet. Availability of a kitchen garden or a cropland, farm animal, remittances, or residence area are associated with higher DDS and micronutrient adequacy. Education and income negatively correlated with DDS and nutrient adequacy.

The many programs of WHO, FAO, and WFP provide knowledge of nutrition principles, but nutrition policy is poorly provided at the state level. This paper employed RDI for Russia and Germany for the lack of similar government-approved recommendations in Kyrgyzstan. The development of an RDI for women in Kyrgyzstan is a fundamental step the government should undertake soon. The basis for the development of nutrition recommendations for specific groups should be performed considering cultural, climatic, historical, economic, and political factors. Altogether, Kyrgyzstan is undergoing a food system transformation. In coup with traditional cuisine, western-type nutrition is becoming common in cities.

Based on the findings mentioned above and discussion, the following recommendations should be considered:Teach household members including women on the organization of nutrition, food storage, planning of food intake for a week, and food preservation training at the household level.Inclusion of nutrition classes in the curricula in kindergartens, schools, and universities. This is vital for establishing healthy eating behaviors for future mothers.Agriculture should be diversified in the mountainous areas of Kyrgyzstan.

Our field research noticed that households tend to concentrate their food production only on several crops. For instance, in the mountainous area of At Bashy, families predominantly grow starchy foods, such as potatoes and barley. While climatic conditions permit the growing of fruits and vegetables, people in the mountainous regions of Kyrgyzstan tend to avoid them, which might likely be due to the lack of experience of planting and developing them, remnants of nomadic food culture, and the current semi-nomadic lifestyle. In comparison to the mountainous area, households in the valley part Aravan grow various fruits, vegetables, and dark green leafy vegetables, such as figs, lemon, dill, parsley, coriander, quince, persimmon, carrots, chives, etc. Diverse food production should lead to diversification of household members’ diets. Therefore, we recommend the diversification of agriculture as a strategy for improving diet quality, especially in mountainous villages.

## Figures and Tables

**Figure 1 nutrients-14-00289-f001:**
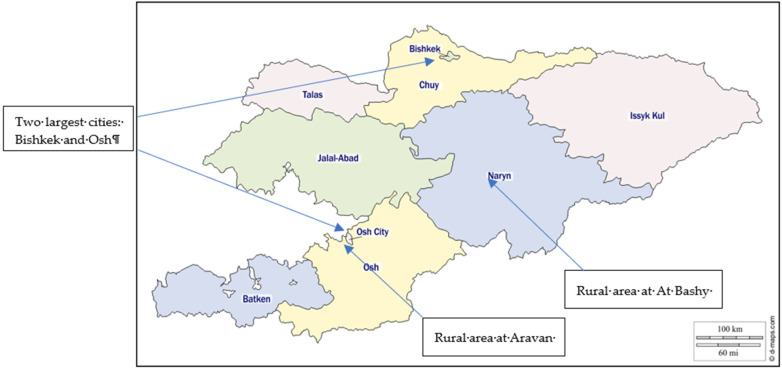
Map of Kyrgyzstan with an indication of field locations [51].

**Figure 2 nutrients-14-00289-f002:**
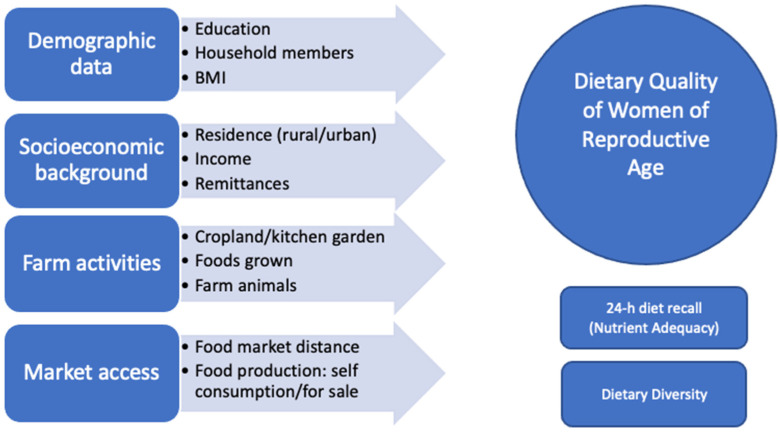
Determining factors impacting dietary quality of WRA (Source: authors’ visualization).

**Figure 3 nutrients-14-00289-f003:**
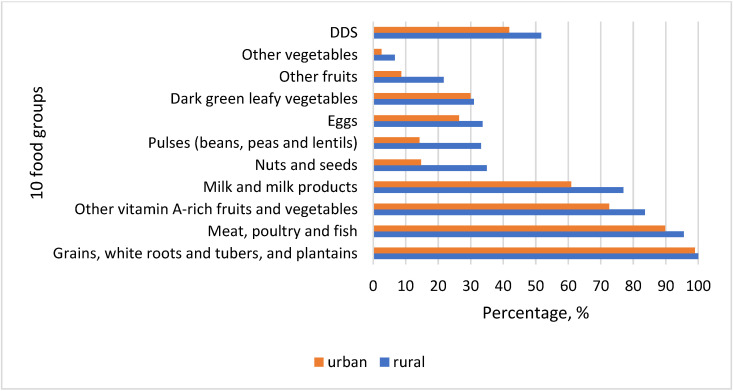
Consumption of ten food groups among women of reproductive age (WRA), and dietary diversity score (DDS) in rural and urban areas.

**Figure 4 nutrients-14-00289-f004:**
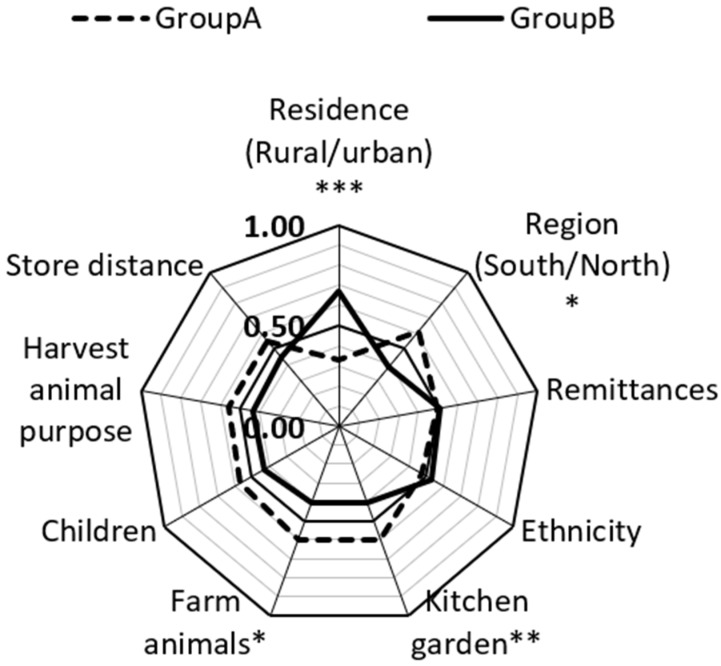
Determining factors for dietary diversity. The figure shows CLES-common language effect size in two groups. **Group A** includes urban residents; -northern region; remittances; non-Kyrgyz ethnicity; kitchen garden, farm animals, children; harvest animal purpose is to mostly self-consume; store distance more than 2 km. **Group B** includes rural residents, southern region; no remittances; Kyrgyz ethnicity, no kitchen garden, farm animals, or children; harvest animal purpose is mostly to sell; store distance-2 km or less. * *p*-value < 0.05; ** *p*-value < 0.001; *** adjusted *p*-value is < 0.001 (The Benjamini–Hochberg method).

**Figure 5 nutrients-14-00289-f005:**
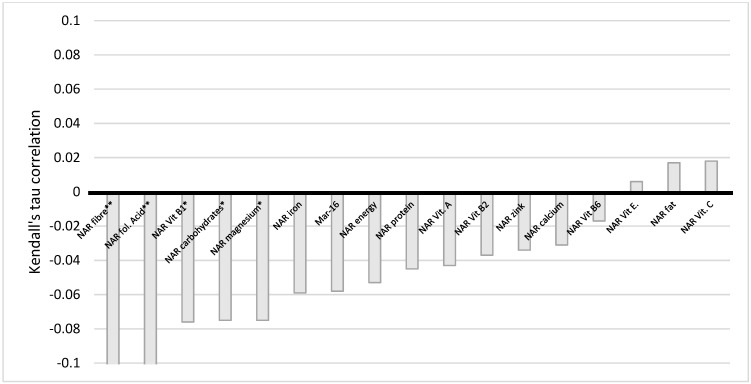
Correlation of NAR for nutrients with income. NAR—nutrient adequacy ratio; Mar-16—mean nutrient adequacy for 16 nutrients * *p*-value < 0.05; ** adjusted *p*-value < 0.05.

**Figure 6 nutrients-14-00289-f006:**
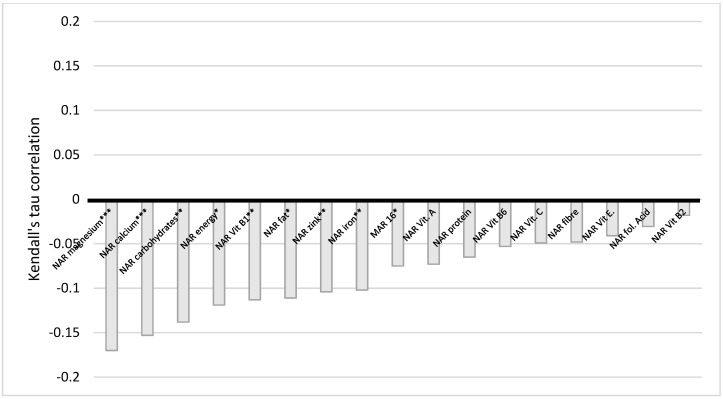
Correlation of NAR for nutrients with education. NAR—nutrient adequacy ratio; Mar-16—mean nutrient adequacy for 16 nutrients, * *p*-value < 0.05; ** adjusted *p*-value < 0.05; *** adjusted *p*-value < 0.001.

**Figure 7 nutrients-14-00289-f007:**
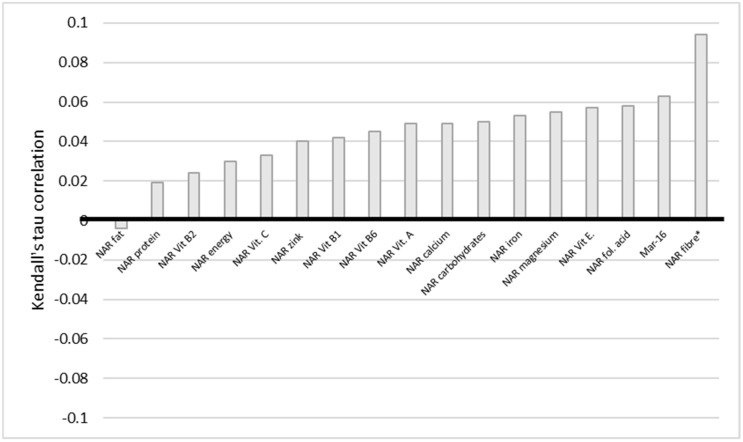
Correlation of NAR for nutrients with BMI. BMI stands for body mass index. NAR—nutrient adequacy ratio; Mar-16—mean nutrient adequacy for 16 nutrients; * *p*-value < 0.05.

**Figure 8 nutrients-14-00289-f008:**
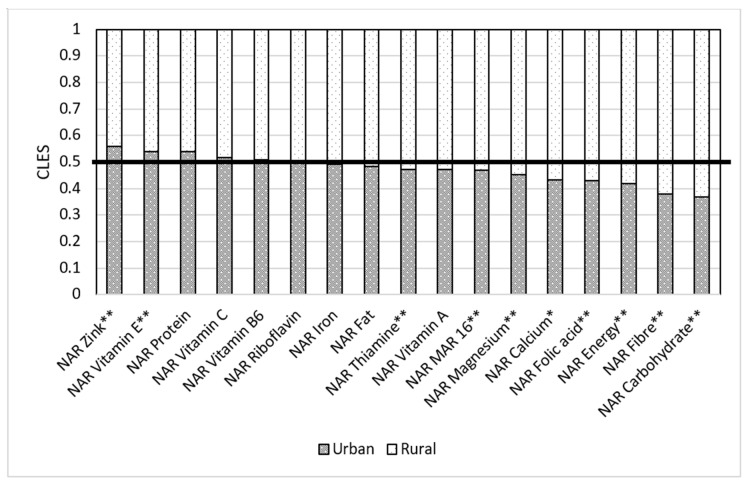
Comparison of NAR for 16 nutrients with urban and rural areas. NAR—nutrient adequacy ratio; Mar-16—mean nutrient adequacy ratio for 16 nutrients; CLES—common language effect size; Wilcoxon *p*-value; * *p*-value < 0.05; **adjusted *p*-value < 0.05.

**Figure 9 nutrients-14-00289-f009:**
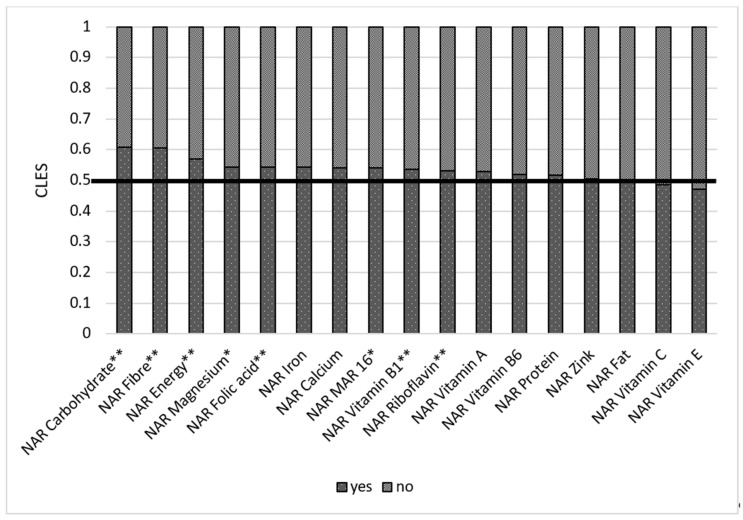
Comparison of NAR for 16 nutrients with the presence of kitchen garden/cropland. NAR—nutrient adequacy ratio; Mar-16—mean nutrient adequacy ratio for 16 nutrients. CLES—common language effect size; Wilcoxon *p*-value; * *p*-value < 0.05; ** adjusted *p*-value < 0.05.

**Figure 10 nutrients-14-00289-f010:**
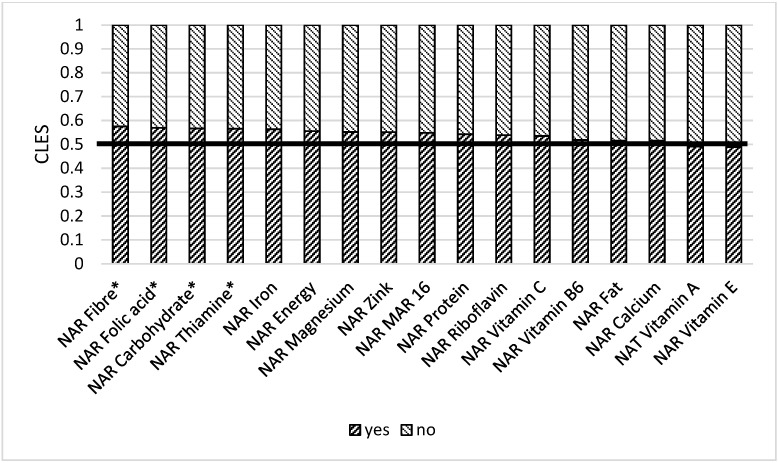
Comparison of NAR for 16 nutrients with getting remittances or not. NAR—nutrient adequacy ratio; MAR 16—mean nutrient adequacy ratio for 16 nutrients. CLES—common language effect size; Wilcoxon *p*-value; * *p*-value < 0.05.

**Figure 11 nutrients-14-00289-f011:**
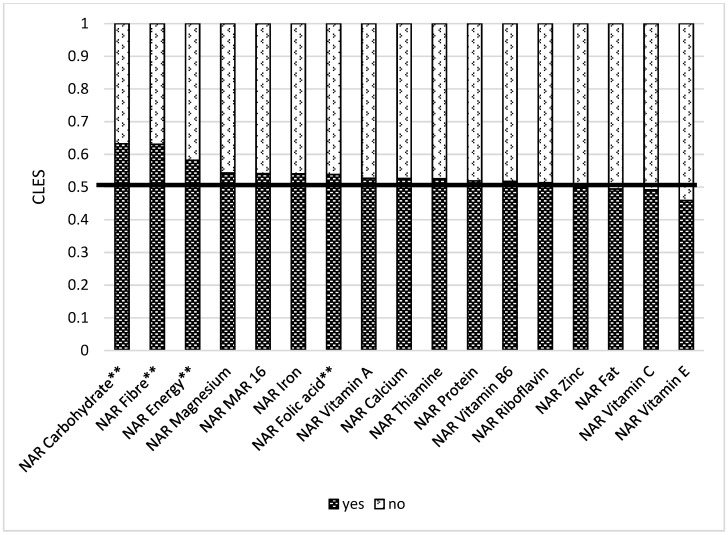
Comparison of NAR 16 nutrients with having at least one farm animal or not. NAR—nutrient adequacy ratio; Mar-16—mean nutrient adequacy ratio for 16 nutrients. CLES—common language effect size; Wilcoxon *p*-value; ** adjusted *p*-value < 0.05.

**Table 1 nutrients-14-00289-t001:** Study area information.

Region	North		South	
**Location**	Bishkek	At Bashi	Osh	Aravan
**Urban/rural**	Urban	Rural	Urban	Rural
**N of respondents**	99	90	98	136
**Geographical coordinates**	42.87′ N 74.59′ E	41.16′ N 75.80′ E	40.52′ N 72.79′ E	40.51′ N 72.49′ E

**Table 2 nutrients-14-00289-t002:** Socioeconomic characteristics of respondents (*n* = 423).

Characteristic	Rural *n* (%)	Urban *n* (%)	Total Value *n* (%)
Age (years):			
18–27	50 (22.1%)	77 (39%)	127 (30.0%)
28–39	96 (42.5%)	66 (33.5%)	162 (38.3%)
40–49	80 (35.4%)	54 (27.4%)	134 (31.7%)
Ethnicity:			
Kyrgyz	105 (46.5%)	182 (92.3%)	287 (67.7%)
Non-Kyrgyz	121 (53.5%)	15 (7.6%)	136 (32.3%)
Education:			
Primary Education	35 (15.5%)	8 (4%)	43 (10.2%)
Secondary Education	104 (46%)	66 (33.5%)	170 (40.2%)
Vocational School	25 (11%)	16 (8.1%)	41 (9.7%)
Higher Education	60 (26.5%)	107 (54.3%)	167 (39.5%)
Marital status:			
Single	29 (12.8%)	66 (33.5%)	95 (22.5%)
Married	189 (83.6%)	120 (60.9%)	309 (73.0%)
Divorced	8 (3.5%)	11 (5.6%)	19 (4.5%)
Household Income (KG soms) *:			
0–5000	32 (14.2%)	21 (10.6%)	53 (12.5%)
5000–10,000	82 (36.3%)	42 (21.3%)	124 (29.3%)
10,001–20,000	67 (29.6%)	66 (33.5%)	133 (31.4%)
20,001–30,000	29 (12.8%)	55 (27.9%)	84 (19.9%)
30,001–40,000	10 (4.4%)	10 (5%)	20 (4.7%)
40,000 and More	6 (2.7%)	3 (1.5%)	9 (2.1%)
Family Size:			
4 or Less	55 (24.3%)	116 (58.9%)	171 (40.4%)
More than 4	171 (75.7%)	81 (41.1%)	252 (59.6%)
Receive remittances:			
Yes	67 (29.6%)	26 (13.2%)	93 (22.0%)
No	159 (70.4%)	171 (86.8%)	330 (78.0%)
BMI level (kg/m^2^):			
Underweight	13 (5.7%)	19 (9.6%)	32 (7.6%)
Normal Weight	118 (52.2%)	118 (59.9%)	236 (55.8%)
Overweight	59 (26.1%)	44 (22.3%)	103 (24.3%)
Obese	36 (15.9%)	16 (8.1%)	52 (12.3%)
Mean (±SD)	24.82 (±4.47)	23.52 (±4.82)	24.21 (±4.68)
Children:			
Yes	175 (77.4%)	98 (49.7%)	273 (64.5%)
No	51 (22.6%)	99 (50.3%)	150 (35.5%)
Source of income:			
Employment	97 (42.9%)	155 (78.7%)	252 (59.6%)
Agriculture	65 (29%)	6 (3%)	71 (16.8%)
Remittances	41 (18.1%)	22 (11.2%)	63 (14.9%)
Private Business	17 (7.5%)	10 (5%)	27 (6.4%)
Government Support	6 (2.6%)	4 (2%)	10 (2.4%)

* KG soms is the local currency, 1 USD = 84.8 KG soms; BMI stands for body mass index; Underweight (below 18.5), normal weight (18.5–24.9), overweight (25.0–29.9), obese (30.0 and above).

**Table 3 nutrients-14-00289-t003:** Food groups and locally consumed foods obtained from 24-hour diet recall.

Food Groups	Reported and Locally Consumed Foods
Grains, white roots and tubers, and plantains	Wheat, rice, buckwheat, pearl barley, wheat bread, pasta products, potato, oatmeal, radish
Meat, poultry and fish	Beef, mutton, goat, chicken, sausage products (sosiska, kolbasa), smoked sausage, horse sausage (chuchuk), liver, stomach, intestine, canned fish
Other Vitamin A-rich fruits and vegetables	Carrot, sweet red pepper, pumpkin, apricot, dried apricot, peach, persimmon, melon
Milk and milk products	Cow milk, ayran, kefir, dried suzmo (kurut), concentrated ayran (suzmo), cream (kajmak, smetana), ghee (sary maj), fermented mare’s milk (kymyz), quark (tvorog), cheese
Nuts and seeds	Walnuts, peanuts, pistachios, sunflower seeds
Pulses (Beans, peas and lentils)	Red beans, chickpeas, mung beans, lentils
Eggs	Chicken eggs
Dark green leafy vegetables	Dill, coriander, parsley, green garlic, green onion, grape leaves, rhubarb, chives
Other fruits	Apple, banana, dried fruits, oranges, cherries, quince, pomegranate, prune, raspberries, sandthorn, currant berries
Other vegetables	Cabbage, garlic, cucumber, tomato, onion, eggplant, beetroot

In parenthesis, Kyrgyz names are indicated.

**Table 4 nutrients-14-00289-t004:** Nutrient intake and its adequacy measured by 24-hour diet recall (*n* = 423).

Nutrient	Rural (*n* = 226)			Urban (*n* = 197)			Total Group			RDI	
	Mean ± SD	NAR Russia	NAR DGE	Mean ± SD	NAR Russia	NAR DGE	Mean ± SD	NAR Russia	NAR DGE	Russia	DGE
Energy (kcal)	1731.42 ± 569.4	0.81	0.79	1551.58 ± 611.26	0.72	0.71	1647.66 ± 595.36	0.77	0.75	2150	2200
Protein (g)	58.14 ± 21.51	0.89	1.12	61.51 ± 30.00	0.95	1.25	59.71 ± 25.84	0.92	1.18	65	50.70
Fat (g)	61.98 ± 31.98	0.86	0.85	59.68 ± 30.89	0.83	0.81	60.91 ± 31.46	0.85	0.83	72	73.3
Carbohydrates (g)	231.57 ± 89.99	0.74	0.84	188.67 ± 86.41	0.61	0.69	211.59 ± 90.80	0.68	0.77	311	275
Fiber (g)	17.99 ± 7.04	0.90	0.60	14.90 ± 6.91	0.74	0.50	16.55 ± 7.14	0.83	0.55	20	30
Vitamin A (µg)	910.09 ± 997.47	1.01	1.30	820.24 ± 665.65	0.91	1.17	868.24 ± 859.24	0.96	1.24	900	700
Vitamin E. (mg)	11.20 ± 8.02	0.75	0.93	12.34 ± 8.94	0.82	1.03	11.73 ± 8.47	0.78	0.98	15	12
Vitamin B1 (mg)	0.62 ± 0.28	0.41	0.62	0.59 ± 0.35	0.39	0.59	0.60 ± 0.31	0.40	0.60	1.5	1
Vitamin B2 (mg)	0.78 ± 0.33	0.43	0.71	0.77 ± 0.33	0.43	0.70	0.77 ± 0.33	0.43	0.70	1.8	1.1
Vitamin B6 (mg)	0.92 ± 0.36	0.46	0.66	0.93 ± 0.43	0.46	0.66	0.92 ± 0.39	0.46	0.66	2	1.4
Fol. acid (µg)	165.2 ± 74.17	0.41	0.55	145.75 ± 79.53	0.36	0.49	156.14 ± 77.24	0.39	0.52	400	300
Vitamin C (mg)	42.01 ± 36.84	0.47	0.44	44.52 ± 45.98	0.49	0.47	43.18 ± 41.31	0.48	0.45	90	95
Calcium (mg)	323.16 ± 161.04	0.32	0.32	286.13 ± 132.01	0.29	0.29	305.92 ± 149.21	0.31	0.31	1000	1000
Magnesium (mg)	198.82 ± 72.59	0.50	0.65	185.24 ± 82.05	0.46	0.61	192.50 ± 77.35	0.48	0.63	400	305
Iron (mg)	10.08 ± 3.61	0.56	0.67	9.93 ± 4.42	0.55	0.66	10.01 ± 4.00	0.56	0.67	18	15
Zinc (mg)	9.25 ± 3.75	0.77	1.16	10.19 ± 5.34	0.85	1.27	9.69 ± 4.58	0.81	1.21	12	8

RDI stands for recommended daily intake; NAR stands for nutrient adequacy ratio; SD is standard deviation; DGE stands for Deutsche Gesellschaft für Ernährung e.V., the German Nutrition Society.

## Data Availability

The data presented in this study are available upon request.

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
