# Peer review of "Dietary Quality of Women of Reproductive Age in Low-Income Settings: A Cross-Sectional Study in Kyrgyzstan"

_nutrients, 2022, doi:10.3390/nu14020289_

Round 1

Reviewer 1 Report

Thank you for giving me the opportunity to review the manuscript of Otunchieva et al.

The work is interesting. Although several elements in this manuscript need to be improved.

Generally

1.The authors use too many abbreviations. And the abbreviation WRA is rather unnecessary.

Abstract

  1. Line 17: Change the "median" to "average value".
  2. The value of the average BMI ±SD is different in the abstract (±4.6) and different in Table 2 (±0.28).
  3. No explanation for "DDS".
  4. No explanation for "MAR 16".

Introduction

  1. A separate, even a short paragraph is needed on the involvement of (presented) vitamins and microelements in cellular processes important for health, e.g. in DNA methylation, antioxidant, anti-inflammatory and immunological activities, and others. These actions are important for a properly developing pregnancy.
  2. At the beginning (again) you should describe all the abbreviations.
  3. Shortcuts are used too many times. It's a good time to use full names sometimes.

Material and Method

  1. Again, a lot of shortcuts.
  2. Line 270: Why was the Mann-Whitney U test used to compare continuous variables?
  3. Are the medians shown somewhere? If not, please write that "medians have been calculated but averages have been shown".
  4. What test compared categorical or dichotomous variables?
  5. What were the definitions of BMI and its categories: how BMI was calculated and what were the ranges for categories.

Results

  1. The abbreviation WRA is definitely overused and it is difficult to understand its use.
  2. Table 2: No units (e.g. for age or BMI).
  3. Table 2: No effect of comparisons between groups. You can even mark those characteristics that differed significantly statistically (e.g. *). After all, you have reported using the Mann-Whitney test. Tests for other variables were also probably used.
  4. Tables: no descriptions under the tables, e.g. no descriptions for abbreviations and for statistical tests.

Discussion

  1. Please re-emphasize the effects of vitamins and microelements important for the development of pregnancy (since women of reproductive age are studied). Selenium was not evaluated in this study, but it should be mentioned that many other micronutrients are also relevant in reproductive age, especially selenium due to its potent antioxidant properties (Rayman et al.; Lewandowska et al.)

Reviewer 2 Report

The manuscript entitled „Dietary quality of women of reproductive age in low-income settings: A cross-sectional study in Kyrgyzstan" presents an interesting issue,  but a lot of areas must be corrected.  In fact, it should be carefully read and checked, and rewrite.
Please see my specific comments below.
The introduction should be organized and shortened, presenting the most important issues related to the study design  (the aim of the study). Fragments to be corrected are marked in the attached file
The manuscript requires rewriting with the exact definition of what the authors want to show and what is  a scientific novelty, apart from the fact that there are very few works on this group of women. 
Paragraph "Novelty of our research" should be moved to the end of the section the discussion as strengths.
In addition, there is no well-defined aim of the study.

Materials and Methods
Important details on participants are lacking. Please provide more information on selection of participants e.g. inclusion/exclusion criteria  for the study. 
Maybe it's better to present it in the diagram - (Flow chart of sample collection).
Moreover, no information on anthropometric measurements,
Whether the measurements were carried out or whether the women self-report this data.
This section should also be organized and the order of the discussed subsections should be corrected

Results
The results should be corrected, too many repetitions of descriptions from tables and figures in the text.
This section does not read well.
Figures 5-7 are unclear to me - in my opinion, this is not how correlations should be presented.

The discussion, in my opinion, should be strengthened and  take into account more publications in this field. Moreover, it should be grouped thematically.
Manuscript and references should be adapted to the editorial requirements.

Round 2

Reviewer 1 Report

Thank you.

I am satisfied with the work done by the authors of this manuscript.

I propose to accept this article.

Reviewer 2 Report

The article has been significantly improved, but tables 2 and 4 are still not readable. Maybe it is worth presenting them on the whole page - widening the columns.
In addition, the literature needs to be improved as required by the journal.
I suggest reading the text carefully before publishing.
